# Evaluation and Optimization of a MOEMS Active Focusing Device

**DOI:** 10.3390/mi12020172

**Published:** 2021-02-09

**Authors:** Ulrich Mescheder, Michael Lootze, Khaled Aljasem

**Affiliations:** 1Department of Mechanical & Medical Engineering, Institute for Microsystems Technology (IMST), Furtwangen University, Robert-Gerwig-Platz 1, 78120 Furtwangen, Germany; lomi@hs-furtwangen.de; 2Associated to the Faculty of Engineering, University of Freiburg, Georges-Köhler-Allee 103, 79110 Freiburg im Breisgau, Germany; 3Z-LASER GmbH, Merzhauser Str. 134, 79100 Freiburg im Breisgau, Germany; aljasem@z-laser.de

**Keywords:** adaptive optics, MOEMS, SOI, electrostatic actuation, active focusing, thin film stress, FEM

## Abstract

In this paper we present a detailed evaluation of a micro-opto-electromechanical system (MOEMS) for active focusing which is realized using an electrostatically deformed thin silicon membrane. The evaluation is done using finite element methods and experimental characterization of the device behavior. The devices are realized in silicon on insulator technology. The influence of internal stress especially resulting from the high compressive buried oxide (BOX) layer is evaluated. Additionally, the effect of stress gradients in the crystalline device layer and of high reflective coatings such as aluminum is discussed. The influence of variations of some important process steps on the device performance is quantified. Finally, practical properties such as focal length control, long-term stability, hysteresis and dynamical response are presented and evaluated. The evaluation proves that the proposed membrane focusing device is suitable for high performance imaging (wavefront errors between λ/5–λ/10) with a large aperture (5 mm).

## 1. Introduction

Active focusing devices allow a dynamic adjustment of the focal length in an optical system. Besides in imaging systems, such devices are used in laser scanning systems, laser micromachining, bar code readers, fiber switches and in projection systems. Technologies for the adjustment of focal length include displacement techniques, e.g., of lenses [1,2] or the use of metasurfaces [3] or the control of fluids (electrowetting) [4]. Another widely used method is the construction of deformable mirrors in the form of membranes (called in this paper FMM: focusing membrane mirrors) that are deformed into a parabolic shape to create a focal point for incident light thus forming reflective lenses where a defined, mostly parabolic membrane deformation can be achieved by electrostatic forces between the membrane as deformable electrode and a suitable, fixed counter electrode. This approach is especially feasible for miniaturization and for fabrication using standard micromachining techniques allowing also high volume production. Different membrane materials have been explored like silicon nitride thin layers [5,6] or polymeric material like SU-8 [7]. An interesting approach is the use of silicon on insulator (SOI) substrates [8]. SOI technology lends itself well to the fabrication of such membranes, since the involved silicon layers, making up the device- and handle-layer, are of single crystal silicon and should therefore exhibit strongly reduced intrinsic stresses compared to materials deposited using physical or chemical vapor deposition (PVD or CVD) thin film techniques. The buried oxide can be used as an etch stop and as a sacrificial layer. Another great advantage is the ability to precisely control the thickness of the device layer, and therefore any structures created from it, over large areas. 

However, even for SOI, residual stresses have to be considered especially for large aperture focusing systems (e.g., with 5 mm active aperture), as high-quality optical properties, specifically optical aberrations of less than λ/5 or λ/10, are increasingly hard to realize reliably. In the realm of smaller apertures many different approaches for focusing systems have already been realized. The role of high compressive stress in the BOX-layer for membrane distortion has been investigated by Sasaki and Hane [9]. For very thin varifocal mirrors (thickness < 1 µm) which were supported at very thin rims (thicknesses 16 and 40 µm and mirror diameter of 400 µm and 1040 µm), they derived an in-plane stress value from analytical and experimental data of about of 47 MPa and 21 MPa (tensile stress) for the investigated rim dimensions. In a later paper, the same authors reported a transition from compressive to tensile in-plane stress at mirror diameters above around 500 µm for the investigated designs [10].

The results presented in this paper are based on the development of stress-free micromirrors with large apertures (5 mm) using SOI wafers as the basis for creating large, free-standing membranes [11]. In that study the correlation between different membrane suspensions and the shape of the counter electrode was investigated, to encourage a parabolic deformation of the membrane itself. This led to the discovery that membranes created from SOI still exhibited buckling of multiple hundred nanometers caused by intrinsic stresses. In a follow-up, we tried to address this problem by designing the membrane suspension with beams running tangentially to the membrane to allow it to deform in-plane and to relax intrinsic stresses without deforming the membrane out of its plane. Using this method, we reported very low membrane buckling of about 48 nm [12]. In [12] we built upon the previous work by creating a fully 3D FEM model of the entire membrane chip, including the surrounding anchor material, in order to predict the behavior of the realized membrane for different beam numbers and different intrinsic stresses in the device layer and the influence of these factors in the parabolic deformation.

In this paper we will first summarize the achieved optical performance characterized at fabricated devices using this concept. Then, we will provide a detailed analysis of that approach using FEM simulation and experimental data. Especially, we will analyze influences of process variations on the performance of the derived devices and provide properties of this focusing device reflecting aspects of specific applications.

## 2. Concept and Design of Electrostatically Deformed Active Focusing Membrane Mirror

The first basic concept of the investigated electrostatically driven FMM for dynamic focusing was presented in [11,13,14]. The mirror device was designed for a nearly perfect parabolic shape when deflected by an electrostatic force. This was achieved using weak beam suspensions in combination with ring-shaped counter electrodes, which provides electrostatic forces especially near to the suspension. Using these ring-shaped counter electrodes, the parabolic range is increased considerably, almost to the full diameter of the used membrane (e.g., diameter of 6 mm). However, residual stress in the BOX and also in the device layer of the membrane can cause buckling and can introduce a convex-like bow of up to several micrometers in its rest position (planar situation, without electrostatic deflection of the membrane, i.e., focal length infinity), which is most sensitive to residual stress in the layers and materials used in the system [15]. To overcome this problem, special formed beams were presented in [12] which allow an in-plane release of residual stress in the membrane material without out-of-plane buckling and thus reducing possible wavefront errors of the focused light.

The basic concept of the FMM investigated in this paper is shown in Figure 1a–c schematically. The mirror is designed as an electrostatically deformable thin plate (membrane) made from crystalline silicon in SOI technology with a typical thickness of 10 µm. The complete device is designed as a hybrid system consisting of a chip manufactured from SOI wafer defining the mirror as deformable membrane electrode and the beam suspensions and a fused silica chip with the counter electrode in an etched cavity providing the fixed counter electrode and the air gap between the mirror electrode and the counter electrode. In principal, two types of counter electrodes can be used: holohedral (full area) electrodes and ring-shaped electrodes. Whereas the former allow relatively low driving voltages, the latter provide a better parabolic shape over a wide membrane area, which is needed for applications with a large aperture [14], and avoid pull-in because the largest mechanical deformation is found in the center of the round membrane and therefore at a position without an opposing counter electrode in the case of ring-shaped counter electrodes. 

Different to previous designs used in [12] the outer diameter of the ring-shaped electrodes in the FMM investigated in this study was designed here so that the ring-shaped electrode ranges up to the rim where the beams are suspending the membrane (space for the tangential beams between the outer edge of the membrane and the inner edge of the rim: 500 µm). The two chips are bonded using point-like adhesive contacts, which is also important for reduction of distortion because the outer, not thinned chip area shows a large bend upwards after release from the wafer, which will also cause distortion in the thin membrane when this chip is bonded to a flat reference plane. The mirror membrane is suspended using silicon beams (8–28 have been investigated here), which are fixed at the silicon rim. Electrical contact to the electrostatically deformed membrane is provided via aluminum interconnects on the SOI device layer. The control of membrane deformation and thus the focal length of the focusing device is achieved using appropriate settings of the driving voltage (typically up to 250 V). The diameter of the active mirror is typically 6 mm, allowing a numerical aperture of about 5 mm (parabolically deformed part of membrane).

As discussed in [12], a very important feature of the design is the use of special tangential beam suspensions of the membrane, which are shown in Figure 2a,b. After release from the substrate, conventional rigidly clamped membranes with intrinsic compressive stress can only expand perpendicular to the surface (out-of-plane buckling). This leads to a kind of Euler’s buckling when a critical load is reached due to the compressive stress, resulting in an out-of-plane movement of the membrane and thus an out-of-plane distortion. With only radial beams (directing towards the center of the membrane) the expansion of the membrane can be transferred to the weaker beams. However, this leads also to an out-of-plane movement of the beams resulting in a shift of the suspended membrane also out of the plane and thus creating wavefront errors when this membrane is used for focusing. A better performance of the focusing device is achieved for beams, which are tangentially fixed to the membrane. Here, the degrees of freedom are extended to the x-y direction (in plane distortion of membrane and beams). This leads to a stress relief and a decrease in the out-of-plane distortion to negligible values corresponding to the beams’ stiffness (wavefront errors less than λ/5 were reported in [12]).

## 3. Methodology

For detailed parameter studies, FEM simulation is used. As much as possible, the simulation results are compared with experimental data taken at specific parameter settings.

### 3.1. FEM Simulation

COMSOL multiphysics was used. In this work we created two basic 3D models to accurately predict the behavior of the mirror membranes. The first “membrane only” model consisted of a thin cylinder of 10 µm thickness to recreate the membrane itself, to which the tangential beams were then added, by modelling a single beam and then rotating it around the center of the membrane. For investigation of the influence of a high reflective coating, an aluminum layer was modelled by repeating the same process with a 250 nm thin membrane and beams, directly on top of the established structure. A second cylinder forms the air between the membrane and counter electrode, which itself is formed of a third cylinder that has a smaller cylinder cut from it (forming a ring-shaped counter electrode).

The physics modules used were the structural mechanics and electrostatic modules, which were connected with each other through the electro mechanics multiphysics module. The flat ends of the beams, normally connected to the surrounding silicon material, were defined as fixed constraints to allow this model to be solved. With this smaller model, we could conduct most simulations to identify and solve fundamental problems early, as well as testing some early model designs without needing to execute time consuming calculations on big and complex models.

In the second model, we included the influence of intrinsic stresses inside the surrounding material (rim) on the membrane and its deformation. To this end, we modelled the membrane as a part of a fully cut chip instead of a separated component. This was realized by first creating an SOI “chip”, which then has the membrane and beam structures added to it. A handle layer, buried oxide and a device layer are created as a block of 10 × 10 mm^2^ area and thicknesses of 380 µm, 1 µm and 10 µm, respectively. Next, a cylinder is cut from that chip to create the cavity, which in reality is etched through the handle and oxide layer. The membrane is then added by creating a cylinder at the same height and with the same thickness as the device layer. To model the beam suspension, an anchor block was added to the membrane edge, from which the beam, running tangentially to the membrane circumference, extrudes into the surrounding device layer. These two components of the beam are now duplicated around the membrane using the “rotate” feature. To make the mesh creation easier, help lines were created on the surface of the device layer. Then, a block of air was added on top of the structure and the outline of the counter electrode created on top of that. Finally, the different blocks that make up the device layer and the membrane are merged to create a continuous domain. The reflective aluminum layers were created by adding the “shell” property offered by COMSOL’s Shell physics module to the corresponding surfaces of the device layer.

For the simulation based on this model, the structural mechanics, electrostatics and shell physics module were used and connected with each other with the electro-mechanics and shell-solid connection multiphasic module. 

In order to define the whole structure in such a way that the chip would be free to deform due to the intrinsic stresses inside the different layers, no fixed constraints could be used. Instead, prescribed displacements were used to forbid certain movements for specific points within the model. For this to be possible, it was assumed that the membrane deforms symmetrically around its center point, i.e., the center point of the membrane was defined to only move along the *z*-axis, forbidding any lateral movement along the *x*- and *y*-axis (equivalent to a sleeve bearing). Additionally, all points inside the surrounding chip material (rim) that lie on the *x*-axis were defined to forbid any movement in y-direction. The same was done with any of the equivalent points on the *y*-axis, forbidding any displacement in x-direction (equivalent to a friction bearing). Finally, the four bottom corner points of the handle layer were prevented from displacing in any z-direction (again equivalent to a friction bearing).

### 3.2. Experimental Characterization

Three-dimensional characterization was done with a white light interferometer (ZYGO NewView7100) with a vertical resolution of 0.1 nm and a horizontal resolution of 0.52 µm or 9.5 µm (50× or 1× objective). Using this tool, we determined the critical dimensions of the mirrors (like distance to the counter electrode, initial buckling and symmetry) and documented their behavior while being actuated through an applied voltage. Because of the relatively large diameter of the membranes (6 mm), we used a special Michelson interferometer objective with a 1x magnification and the stitching function of the Zygo to document the distortion (at U = 0 V) or the intended deformation (U > 0 V) of the whole chip surface.

In order to actuate the mirror systems, we utilized a standard DC generator and a small DC piezo amplifier (Piezo Systems, Inc.; Proportional Voltage Booster, Cambridge, MA 02139, USA) to create voltages of up to 275 V and membrane mirrors, already placed into their housing. The voltage was increased and decreased in 25 V-steps, typically from 0 V to 275 V. After each voltage the variation of the surface profile of the membrane or chip was measured. 

By using the junction of beams and surrounding chip as a reference, we determine the displacement of the membrane, extracting both the characteristic deformation behavior of multiple mirrors to compare to the simulations, and their hysteresis. The maximum deformation w_0_, also defining the parabolic radius at a given voltage, was derived as the difference of the measured membrane deflection in the center and at the membrane rim.

### 3.3. Fabrication

For the realization of the micromechanical components (deformable mirror and tangential beam suspensions), double side polished 4” SOI wafers were used. The counter electrode and the cavity defining the electrode distance are made out of 4” fused silica wafers. The fabrication of the mirror systems was executed in three separate stages: the membrane chip, the glass counter electrode and the packaging. 

#### 3.3.1. SOI-Membrane Chip

As the basis for the membrane chip, SOI wafers from IceMOS Technology (Belfast, UK) (results in Section 4.1) and Si-Mat-Silicon Materials e.K. (Kaufering, Germany) (results in Section 4.2) were used. The thicknesses of the device and handle layer and BOX layer are shown in Table 1. First an aluminum film of about 250 nm thickness was deposited on the device layer side of the wafer via thermal evaporation. This film was then structured through contact lithography and in a wet etching step to define the beams and membrane structures. The structured aluminum layer was then used as a mask to shield the silicon against the following reactive ion etching (RIE) step, which transferred the structures into the device layer. The buried oxide (BOX) served as an etch stop. After that, the remaining aluminum on the device layer side was again structured via photolithography and wet etching, to remove all aluminum aside from small contact pads. The device layer side was then coated with a thick layer of photoresist, which served as a protective layer. The backside handle layer was then structured with contact lithography in conjunction with IR-backside alignment, to define the membrane radius from the back side. The silicon of the handle layer was then removed through deep reactive ion etching (DRIE). The BOX served again as an etch stop. To free the membrane fully, the wafer was submerged into concentrated HF acid for about 15 min to remove the buried oxide layer. Finally, the photoresist was removed in a plasma incinerator, with a reduced gas flow to prevent damage to the membranes. The mechanically sensitive membrane chips where then separated from the wafer with a nanosecond cutting laser (LS 9000, Laser Systems GmbH, Krailling, Germany). More details including the flow chart used are provided in [12].

#### 3.3.2. Glass Counter Electrode

The glass counter electrode was created from fused silica wafers from Siegert Wafer. The material was chosen both for its transparency to visible and UV light (allowing UV curing of adhesive), as well as for its isolating electrical properties. First the wafers were coated with a 1.5 µm thick layer of polysilicon, which serves as a masking layer. This layer was then structured via photolithography and RIE. Then, 70 µm deep electrode cavities were etched into the glass using wet etching in concentrated HF for 1 h. The remaining poly-silicon was removed in KOH. A thin film of aluminum (250 nm) was deposited onto the glass through thermal evaporation followed by a lithography step. During the application of the photoresist, all cavities were filled before the resist was thrown off via spinning. The following photolithography step and wet etching process structured the counter electrodes into the aluminum. The counter electrode chips were separated off the wafer with a wafer dicing saw.

#### 3.3.3. Packaging

The two chips where bonded together with a Tresky 3002 die bonder using a UV cured adhesive (Norland Products, Optical Adhesive 61, Cranbury, NJ, USA). Average adhesive thickness was 35 µm (±7 µm). As shown in Figure 3a, the SOI chip with the deformable mirror is bonded upside down to the fused silicon chip which contains the counter electrode at the bottom of the etched cavity. Typical dimensions as well as geometries used later are also shown. The electrical interconnect from the device layer to the contact pads on the fused silicon chip was established by locally melting the aluminum pad through the glass with a nanosecond laser (again LS 9000, Laser Systems GmbH, Krailling, Germany). The completed mirror element was finally glued into a constantan housing. Electrical contact to the output pins was established via wire bonding. Figure 3b shows the tangential beams attaching the membrane to the thick rim of the SOI chip. Figure 3c shows a complete system prior to packaging in relation to a match.

## 4. Results

### 4.1. Proof of Concept: System Behavior and Optical Performance

First, we will summarize previously published results which demonstrated the optical performance of the concept for high imaging quality (wavefront errors < λ/5) and large aperture.

The basic function is demonstrated in Figure 4 with results obtained using COMSOL simulation (left) and measurements (right). The electrostatic forces do not only introduce the intended parabolic membrane deformation, but will also pull down the whole membrane because of the weak suspension of the membrane to the Si-rim. Typically, the usable membrane deformation is about 50% of the total electrostatic displacement. However, the relation between down bending of the beams and parabolic deformation of the membrane depends also slightly on electric field size (change by around 5% in the used range of electric field size). This will affect the theoretically expected square dependence of deformation amplitude on electric field and has to be considered in the control electronics and will be discussed further in Section 4.2.2 and Section 4.3.

The measured membrane deformation at the same voltage as the simulated one (Figure 4, right) corresponds well to the simulation and proves the concept of parabolically deformed FMM.

The performance of the FMM for optical imaging has been demonstrated with an optical set-up where collimation optics was used to create a plane wavefront on an iris diaphragm which was then focused on a CCD camera for beam analyses [16]. The according results are shown in Figure 5. Even for an incidence angle of light of 45° (no beam splitter was used), the profile fits well to a perfect Gaussian distribution for an aperture of 2 mm (Figure 5a)). For an aperture of 5 mm (Figure 5b), slight light satellites were observed, which are partially also a result of the non-vertical incidence of light. In [16] it was also shown that the performance of the focusing membrane mirror can be well described by the deformation measurements which were used as input to calculate the Zernike coefficients (terms 1–36). The ZEMAX simulation of the focused laser beam profile using the Zernike coefficients agreed very well with the measured beam profile. Finally, in Figure 5c the measured beam waist as function of focus length f is plotted and compared to the beam waist expected for an ideal parabolic mirror. This demonstrates the large optical dynamic range of the membrane focusing device (f: 200–1000 mm, 1–5 diopter). These results also confirmed that the deformation shape measured with the white light interferometer is a quality measure for the optical performance of the FMM. Therefore, we present membrane deformation data for describing the optical quality of the membranes.

The focal length is also shown in Figure 6. The focal length f is correlated to the maximum deflection w_0_ of the membrane, which occurs under electrostatic deformation in the center of the membrane and to aperture A:(1)$$f=\;\frac{A}{16\cdot w_{0}}$$

The results show the feasibility of the presented concept which is based on an electrostatically-deformed silicon membrane, tangential beams and ring-shaped counter electrodes for varifocal imaging in high aperture imaging systems. However, for successful transfer of the research results to industrial applications, influences which might deteriorate the performance will be investigated in detail in the next sections.

### 4.2. Investigation of Influences of Fabrication and Material Aspects on System Behavior

#### 4.2.1. Distortions Introduced by Mechanical Stress

##### Stress Resulting from SOI

The devices investigated in this section were made out of SOI wafers from Si-Mat, where silicon fusion bonding (FSB) is used for SOI fabrication. The stress situation for MOEMS devices with very thin device layer thickness (<1 µm) and also a thin rim (<40 µm) was related in [10] to large compressive stress within the BOX layer. In this section we will first analyze the stress situation for the presented layout, where a relatively thick (10 µm) device layer and a very thick and wide rim (410 µm and 1.5 mm) are used.

Figure 7 shows a typical result of a mirror chip after cutting that chip out of the SOI wafer using laser dicing. The total chip size is 10 × 10 mm^2^, and the membrane diameter is 6 mm. A design with 24 tangential beams is presented. Obviously, a big part of the total out-of-plane distortion is observed at the rim (here around half of the total up-bending of around 1.5 µm is assigned to the thick rim and the beams).

As a comparison, the results of the COMSOL simulation are shown in Figure 8. Here, constant intrinsic stress values in the device layer ranging from −10 MPa (compressive) to +10 MPa (tensile) were assumed. Even though qualitatively similar to the measured data, it is obvious that the observed membrane distortion shown in Figure 7 cannot be explained by a uniform device layer stress as the membrane distortion in this case is only slightly changing with device layer stress (Figure 8b: around 60 nm/20 MPa). Similar results are found for 12 beams and the same total beam stiffness.

The low simulated distortion (Figure 8b) corresponds to the experimental results presented in [12,16] where the distortion of the mirror itself was very low over the membrane area (<100 nm). However, Figure 7 shows a pronounced convex distortion of the membrane at U = 0 V. This behavior can be explained by a stress gradient within the device layer as a uniform intrinsic stress would be compensated mostly by the tangential beams as shown in [12].

Due to this finding, we analyzed the membrane distortion in detail. Different devices with different numbers of beams (12 and 24) and made from different wafers showed a similar behavior as presented in Figure 9. The total beam stiffness of the evaluated suspensions is comparable as the width of the beams was adjusted so that the product (number of beams × width of beams) is constant. In Figure 9, the distortion between two beam suspension points placed diagonally on the inner radius of the thick rim is shown. The average membrane distortion defined as z-difference between the rim and the center of the according membranes is 1.29 µm (±0.4 µm). No significant dependence on the number of beams is found as the total stiffness is similar. This large out-of-plane distortion can only be explained by a stress gradient within the device layer because a predominant part of a uniform device layer stress is transferred by the tangential beams into an in-plane distortion (Figure 2b), thus considerably reducing the out-of-plane distortion by due to uniform stress.

To prove the thesis that a stress gradient within the device layer can cause the observed membrane distortion, a COMSOL model was built, where the device layer is split into two layers with different intrinsic stress values. The results are shown for a model fitting to the design of the measured chip mem08 (24 transversal beams, red dashed curve in Figure 9). The chosen layer thickness and stress values were 9 µm/−10 MPa and 1 µm/−8.5 MPa, respectively. The COMSOL simulation fits well within the central part of the membrane up to a diameter of 4 mm. However, it should be noted that also other combinations of thicknesses and stress values can be used for reasonable fitting to the experimental data.

The results show that large convex distortion can be related to a stress gradient within the device layer. Additionally, a comparison of results obtained with SOI wafers from different suppliers indicates that also the process of how the SOI wafers are produced has to be considered as a factor influencing the final device performance.

Prior to adhesive bonding, most of the chips showed a point symmetric distortion as shown in Figure 9. However, sometime non-point-symmetric deformations with a fold-shaped surface were found, where the fold structure was directing to a beam suspension. This phenomenon is related to maloperation of specific beams as shown by experimental and simulation results as well. As the folded structure is a kind of bi-stable phase to which the system is switching during the release process, such non-point symmetric distortion will also influence the system after bonding and deteriorate the final system behavior even when electrostatically deformed in a controlled way. Therefore, chips showing such a fold-shaped characteristics are not suitable for high performance imaging (wavefront distortions < λ/5) and were discarded for quality reasons.

##### Influence of Aluminum Reflective Coating

The measured low wavefront error demonstrated in Figure 4 was obtained with Si-membranes without any coating. The reflectivity in air (n(air) = 1) can be approximated by Equation (2) in case of very low absorption, where n is the refractive index of Si at the used wavelength. With n(Si) = 4.0 (λ = 600 nm) a reflectivity of about 36% is obtained.
(2)$$R=\;\frac{\left(n-1\right)2}{\left(n+1\right)2}$$

The reflectivity of the mirror can be increased to about 92% by coating the membrane with a thin layer of aluminum (Al). However, Al deposited with PVD (physical vapor deposition) has high intrinsic stress and will therefore introduce further distortions into the system. To derive the influence of Al on the device performance, first the stress in PVD-deposited Al was investigated via special micromechanical stress indicating structures such as beam and bridge structures. The chosen Al thickness was 50 nm, which was proven as the minimum thickness for high reflective coatings. The experimentally measured deflections were evaluated through COMSOL simulation and Stoney’s equation to obtain the according stress values. For as-deposited PVD aluminum layers, a tensile stress of around 10 MPa was determined. Under annealing up to 300 °C the tensile stress increases with an almost linear dependence of 100 MPa/(ΔT = 100 K). Based on these experimental results, a tensile stress of 10 MPa was chosen as the minimum intrinsic stress of Al for simulation.

Figure 10 shows the COMSOL results for an Al-layer deposited on the surface of the Si-membrane pointing into the handle layer cavity. In this first model including reflective coatings, the Al-layer was modelled as a 500 nm thick layer. To enable meshing, the 1 µm thick BOX-layer was split into two separate layers of 500 nm. This prevents meshing errors caused by unaligned block edges and allows the COMSOL meshing algorithm to succeed. A tensile Al stress is further increasing the membrane distortion already resulting from a compressive device layer stress by several µm. Due to the Al-induced stress, mainly the membrane itself is distorted. The membrane distortion via Al stress is increasing almost linearly with Al stress, at least within the investigated range of Al stress values (Figure 10b).

As a result we conclude that even with the lowest as-deposited Al stress of 10 MPa, the resulting large membrane distortion will prohibit a reasonable use of the device for imaging, especially as any thermal treatment within in the process will further increase the Al stress.

To overcome this limitation for applications which need high reflective FMM, a double side deposition of aluminum was investigated. To prevent computation errors due to high aspect ratios of element dimensions during simulation of these very thin layers, the mesh either had to be very fine, which would increase the computation time greatly, or the layers had to be modelled with a different technique. To this end, we made use of the shell physics module offered by COMSOL, to model the thin aluminum layers not as 3D objects, but as a number of material properties and forces which are attached to the interfaces where the aluminum layers are in contact with the silicon membrane. The results in Figure 11 demonstrate that the same distortion is achieved for double side deposited Al as without Al deposition if the Al stress value and the thickness are exactly equal for both layers. However, normal PVD deposition processes do not provide the needed process control and process stability to make use of the double side deposition approach.

#### 4.2.2. Influence of Process Variations

As for every device, variations in fabrication conditions will also have an influence on the characteristics of the electrostatically deformed FMM device presented here. As the device layer thickness is well defined when using SOI wafers and the width of the tangential beams is defined via photolithography and deep reactive ion etching (DRIE), we can neglect these very small variations of these processes. The most critical influences on the performance of the presented FMM are related to the bonding process of the mirror chip to the fused silica chip containing the counter electrode within the cavity and to the definition of the electrode gap via cavity etching into the fused silicon. In this section, these influences of the process variations on the device performance are investigated.

##### Influence of Tilt between Electrode Surfaces

As discussed in Section 4.2.1, chip based bonding techniques using soft adhesives are most appropriate due to the large warp of the thick rim (Figure 7 and Figure 8). However, a disadvantage of adhesive bonding compared to direct bonding such as anodic bonding or fusion bonding is that the gap between the Si-membrane chip and the fused silicon counter electrode chip is not well defined, especially for the relatively thick adhesives with large viscosity used here. The tilt between the two electrode surfaces, i.e., the surface of the mirror chip and the counter electrode chip, was measured with five different devices by measuring the distances of three different points on the fixed rim, which form an isosceles triangle with 7 mm and 4.95 mm distance. Typically, the measured tilt angles are well below 0.112° and show a relatively large spread, which is associated with the large spread of adhesive thickness (average thickness 35 µm, ±7 µm). Based on the measurements, 0.112° is used as a reference value for the maximal tilt introduced by adhesive bonding. A tilt between the surface will cause an according change of electrode distance and thus of electric field. This might introduce an unbalance to the electrostatic deformation of the mirror membrane compared to the point-symmetric deformation without any tilt. To prove the influence of the measured tilt on the intended deformation at a given electric field, COMSOL simulations were performed using a model with 24 beams and a tilt between the two electrode surfaces. The results are shown in Figure 12 for a layout with 12 tangential beams.

For the experimentally found tilt angles (<0.112°), the effect on the deformation shape is observable, but small. Especially, the simulated deformation shapes can still be fitted to a parabolic shape. However, a direct comparison between 0° (no tilt) and the largest observed angle addressed to adhesive bonding (0.112°) shows maximal differences of about 0.35 µm around the maximum deformation w_0_ of about 12 µm (3%). Even though this effect can be compensated for using optical alignment, as long as the membrane surface is still deformed parabolically, we have determined the maximum tilt angle for which the deformation change is less than λ/10. As shown in Figure 12 (right), this is the case for tilt angles <0.02°. Such an angle corresponds to a height difference of 3.5 µm over a distance of 10 mm (chip size) and defines therefore a quality criterion for the needed thickness uniformity of the used adhesive. In our case, 3.5 µm corresponds to a thickness uniformity of better than 10%, which is a reasonable requirement for the bonding process, especially when using thinner adhesives with lower viscosity.

##### Lateral Shift of Counter Electrode and Membrane Electrode

As the device chip and the counter electrode chip are bonded individually chip by chip, the influence of a relative lateral shift of the two chip and thus a lateral relative shift of the two electrodes should be considered. The presented results are calculated for a model with 12 tangential beams (width 100 µm), a ring-shaped electrode (R_i_ = 1.5 mm, R_o_ = 3.5 mm), a device layer stress of −2 MPa and an applied voltage of 150 V, however, similar results are obtained with other configurations. As described in Section 3, Figure 3a, the geometry of the ring-shaped electrodes is chosen such that the outer electrode radius on the electrode chip is placed, in the case of correct alignment, just opposite to the inner chip rim, i.e., the ring-shaped counter electrode is overlapping the area for the beams between the outer membrane rim and the inner chip rim. Therefore, a shift of 0.5 mm means that the electrode is already on one side below the chip rim and on the other side just opposite to the outer membrane rim, which obviously has a severe influence on the symmetry of the electrostatic forces.

The influence of a relative lateral shift along the *x*-axis on the membrane distortion is shown in Figure 13. For better comparison, the membrane deformations are levelled to the left rim of the membranes. As expected, a lateral shift of the electrodes will introduce an according shift of the membrane deformation and therefore a pronounced deviation to parabolic fits (Figure 13, right). An alignment accuracy much better than 0.5 mm is needed to keep the distortion reasonably low. However, as a die bonder with a lateral positioning accuracy of 10 µm was used for bonding the device chip to the counter electrode chip, the misalignment effect can be neglected.

##### Electrode Distance

The effective electrode distance (gap between electrodes) is defined by two independent process steps: the wet chemical etching with 50% HF (etch rate of 1 µm/min) and the thickness of the adhesive. Whereas the later had a typical variation of ± 7 µm for the devices fabricated for this report, the low etch rate of fused silica allows a tight control by etch time. Additionally, the achieved etch depth can be well monitored during etching.

In a first approximation, the electrostatic force between the ring-shaped electrode can be described using Equation (3).
(3)$$F_{es}=\frac{1}{2}\varepsilon_{0}\varepsilon_{r}U^{2}\pi \left(R_{o}-R_{i}\right)/d2$$
where ε_0_ and ε_r_ are the absolute and relative dielectric constants, U is the applied voltage, R_o_ and R_i_ the outer and the inner radius of the ring-shaped counter electrode and d the distance between the electrodes at U = 0, respectively. As the maximal deformation w_0_ in the center of the membrane is proportional to the electrostatic pressure provided by F_es_, the same dependence on U and d is expected for w_0_. However, both the dependence on voltage U and distance d follow a more complex function, as discussed in [14]. In fact, the dependence on d depends on the (zero volt) distortion, the deformation itself and also on the beam and ring electrode design, e.g., the relation between z-deflection of the week beams and membrane deformation (s. Figure 4 and Figure 8) depends on the beam design as well on the driving voltage. This complex relation can be seen in Figure 14, where the percentage of the membrane deformation relative to the total deformation (beams + membrane) is plotted as a function of applied voltage. Two COMSOL models were simulated: 12 and 24 tangential beams, uniform device layer stress of −2 MPa, ring-shaped counter electrodes (R_i_ = 1500 µm, R_o_ = 3500 µm). Whereas for large voltages the percentage slightly decreases with increasing voltage, the percentage at low voltages is also influenced by the starting deformation (U = 0V). From these results we derive that, e.g., the voltage dependence does not follow the simple approximation in Equation (3), as this approximation does not consider the described effect of stress redistribution between the beams and membrane.

The simulation results for a COMSOL model with 12 beams (width 100 µm), effective uniform stress in the device layer of −2 MPa and a ring-shaped counter electrode with R_i_ = 1500 µm and R_o_ = 3500 µm are summarized in Table 2. The maximum deformation w_0_ was calculated as the difference between the membrane rim and membrane center, thereby only considering the optically active part of the deformation (parabolically deformed membrane).

Even though following approximately the dependence on U and d as in Equation (3) from the values in Table 2, different, interesting findings can be derived for w_0_ within the two parameter space defined by U and d:-The voltage dependence of w_o_ does not follow exactly a U^2^ dependence: the relative deformation w_0_ (150 V)/w_o_(50 V) is only 7.9 for d = 80 µm and 6.9 for d = 40 µm (from Equation (3) 9.0 is expected). Assuming a U^x^-dependence (i.e., neglecting any other terms) results in x = 1.88 at d = 80 µm and x = 1.76 at d = 40 µm-In addition, the dependence of w_o_ on d does not follow exactly a d^−2^ relation: the relative deformation w_0_ (80 µm)/w_o_(40 µm) is only 3.6 for U = 50 V and 3.1 for U = 150 V (from Equation (3) 4.0 is expected). Again, assuming a d^x^ dependence (i.e., neglecting any other terms) results in x = −1.63 at U = 150 V and x = −1.85 at U = 50 V.

Therefore, considering the variation of adhesive thickness as the dominating process influence we expect from the experimentally determined thickness variation of ± 7 µm at a typical distance d of 70 µm (±10% distance variation) a corresponding variation of deformation w_0_ and thus focal length of the membrane focusing device of around ± 17% (average between x= −1.63 and x= −1.85), which can be compensated by an according adjustment of the driving voltage, especially as pull-in effects are considerably reduced by the ring-shaped electrode design. Even though the exact variation of w_0_ for a given distance variation will depend on the specific voltage U and the specific electrode distance d as can be derived from Table 3, the evaluation shows that a reasonable control of optical power is possible even for a large electrode distance variation of 10%.

### 4.3. Application Aspects

Finally, properties related to application aspects are experimentally determined. Several devices with different designs (in respect to number of tangential beams) from two different wafers were characterized.

#### 4.3.1. Focal Length Control

Assuming a parabolic membrane deformation under electrostatic actuation and using Equation (1), the focal length is directly related to the membrane deflection w_0_ measured from the rim of aperture to the membrane center. In this study, we assume that the membrane rim is defining the maximum aperture, however, a smaller aperture can be realized by using appropriate iris diaphragms. From a practical perspective, the focal length for a given device can be controlled by the applied voltage, which is automatically set in an autofocus system where an external control signal for proper focus setting is used. However, even for such an application it is interesting how large the control range has to be due to variations of the parameters influencing the w_0_ (Section 4.2).

The voltage dependence of w_0_ is plotted in Figure 15 for different devices with different designs (12 and 24 beams). A relatively large spread is found for the voltage characteristic especially at large driving voltages. Besides the already discussed influence of the thickness of the used adhesive layer (±10% resulting in around ±17% change of w_o_ at given voltage), also the design parameters and influences of the pre-distortion and the chip-based bonding process will contribute to the total spread in sensitivity Δf/ΔV~Δw_o_/ΔV; e.g., at U = 250 V the experimentally found sensitivity range shows a variation of about 30% in relation to the maximum sensitivity measured. Such a spread will accordingly lower the focusing range (optical power in dpt) of a given device as the maximum voltage which can be used to compensate a low sensitivity is finally limited by pull-in or electrical breakthrough voltage.

#### 4.3.2. Long-Term Creeping

In order to ascertain the stability of the membrane deformation, we actuated the membrane with an AC voltage to study the influence of many focusing cycles. We used a function generator to create a 50 Hz sinus signal with V_pp_ = 70 V, which was the maximum AC voltage of the amplifier. The measurements were done after a specific cycle time again under DC voltage. To minimize the possible detrimental effect of the back and forth switching on the measurements, a second measuring run was conducted, where the membrane was measured at the beginning and then left to oscillate for about 1.08 million times, before being switched back to a DC signal for measurement. The results are presented in Table 3 and Figure 16. For the first test (up to 6000 cycles) the deformation measured after each cycle does not change within the measurement accuracy. For up to 6000 cycles (Figure 16a) the fits show a small increase of 1 × 10^−6^ µm/(cycle no) and 5 × 10^−7^ µm/(cycle no) for the membrane deformation itself and for the deformation including the beams, respectively. However, the data are within the measurement accuracy. Therefore, a second test was made with one million cycles. Only the deformation before stress cycles and after one million cycles was measured to avoid any influence of handling. The data in Figure 16b and Table 3 show a small change, mainly in the total chip deformation (which includes the beams), however, the fitted slopes are rather small: the fits show a small increase of 2 × 10^−7^ µm/(cycle) and 3 × 10^−7^µm/(cycle) for membrane deformation only and deformation including the beams, respectively. However, these values also include other influences such as long-term change of temperature or humidity.

#### 4.3.3. Hysteresis

Hysteresis effects were studied by cycling the voltage four times up and down and measuring the membrane deformation at different voltages within each up- and down-cycle.

Typical results are shown in Figure 17. Within the measurement accuracy no systematic hysteresis could be detected. A few unsystematic steps were attributed to wrong voltage settings, which were done manually in these experiments.

#### 4.3.4. Dynamic Response

The dynamic behavior was investigated using a special optical set-up where a collimator laser beam was focused using a combination of a static lens and the adaptable FMM through an iris diaphragm (diameter 1 mm) on a detector. Without voltage (flat membrane) the focus was set exactly to the iris diaphragm plane. Therefore, maximum detector output is achieved at U = 0V. By applying a voltage to the membrane focusing device, the laser spot at the iris diaphragm is defocused, thus the detector signal is decreasing (the larger the defocused spot size, the lower the detected light behind the iris diaphragm). The size of the detected modulation amplitude was considered as criteria if the mirror could follow the driving control potential accordingly when increasing the frequency of the modulated control potential. To test the dynamic response of the focusing mirror, a square wave voltage between 0 V and 170 V was used. The investigated frequency range was from 100 Hz to 1400 Hz. The results are shown in Figure 18.

The modulation amplitude (maximum-minimum of detector output voltage) is constant up to voltage modulation frequencies of about 300 Hz and decreases by about 40% up to 1400 Hz (Figure 18c). This behavior can be modelled with a RC-model where the calculated capacitance of the used device (R_i_ = 1500 µm, R_o_ = 3500 µm, d = 100 µm) of 2.8 × 10^−12^ F is used and a resistor of 10^7^ Ω (low doped Si) is adjusted as a free fit parameter. Thus, the behavior can be related to a decreasing voltage drop at the capacitor like an FMM device which is electrically in series with the resistor defined by the interconnects (Al) and the plate (Si), (Equation (4)). The corresponding fit is shown in Figure 18c.
(4)Ud= 8,52 Vω·C·R2+(1ω2C2)
where U_d_ is the photodetector output related to the focal length and thus the driving voltage U_c_ which is described as frequency depending term on the right side of Equation (4). ω=2πf is the angular frequency for the given frequency f. As C has been calculated for the given device, R and the numerator value 8.52 V (considering the relation between driving voltage U_c_ and photodetector output U_d_) are the two free fit parameters used to fit Equation (4) to the measured detector modulation.

Additionally, the time dependence of the decreasing detector signal (related to the charging phase of the capacitor defining the FMM) and the increasing detector signal (related to the discharging phase to refocus the laser spot again in the plane of the iris diaphragm) is evaluated in Figure 18b. The two parts of the output signal are plotted with a time shift to get a direct comparison within the same time scale. Whereas the charging time is rather long (1.36 ms, red curve), the discharging time is only 0.2 ms (black curve in Figure 18b), which corresponds exactly to the RC time calculated with the used values for R and C. Therefore, for the charging sequence the time dependence is mainly given by the limited available current provided by the amplifier electronics, which increases the practical charging time. However, the discharging time is given by the shorter RC time τ=R·C. This finding corresponds also very well to the mechanical resonance frequency of f_00_ = 1557 Hz derived from modal analysis in COMSOL. Thus, up to 1500 Hz the electronic circuitry is the limiting factor for the dynamic use of the FMM device, however, at larger frequencies the mechanical resonance and the damping will limit the useful frequency range.

## 5. Discussion and Conclusions

The optical quality of the presented membrane-based active focusing device is especially depending on internal and external stress acting on the device which will introduce distortions into the flexible membrane. The MOEMS devices are realized via SOI technology. In principle, the use of SOI wafers provides three major advantages:-an exact control of the membrane thickness t and thus an exact control of device sensitivity at a given driving voltage (t^−3^-dependance of membrane stiffness) provided by the thickness of the device layer and the defined etch stop at the BOX when defining the membrane from the backside,-the mirror and the beams are out of crystalline Si (device layer) thus providing low intrinsic stress (theoretically zero stress) compared to thin films deposited using PVD or CVD techniques and avoiding any creeping or fatiguing of the mirror deformation related to the membrane itself,-simple fabrication of the MEMS components (for the device chip presented here only two lithography steps were needed).

However, even assuming a crystalline Si layer, the large stress in the BOX (we assumed a compressive stress value of −300 MPa as reported in [17]) will introduce stress in the device layer as described in [9,10], where after definition of the mirror thickness via etching of the handle layer and the BOX from the back side of the rim, the mirror is warped at a certain angle due to a circumferential bending moment. From these findings it can be derived that it is very important to consider not only the membrane itself but also the (thick) rim of the mirror chip to understand the stress situation and thus the distortion of the final device. We also found that SOI wafers from different suppliers show different effects. Especially the stress and stress gradients in the device layer of the SOI are crucial and need to be specified or characterized.

The analysis of stress introduced by typical high reflective coating such as PVD deposited aluminum showed that due the big influence of the stress in the coating on the membrane distortion, a standard metallization is not feasible for such FMM devices. New reflective coatings such as metal-organic vapor-phase epitaxy (MOVPE) multilayers are possible candidates to overcome this limitation.

The pronounced upward bending of the thick rim of the device chip indicates that the final distortion will be influenced by the later bonding process. Whereas for direct bonding techniques such as anodic bonding the bonded chip surface is defined by the substrate surface to which the chip or wafer is bonded to, bonding via soft adhesive will allow the pre-distorted surface plane of the mirror chip to be preserved and thus will avoid stress relaxation within the beams or be even more harmful for the final performance stress relaxation within the reflective membrane. Therefore, in this work chip-based adhesive bonding was used. This provides on one side a soft bonding of the warped device layer chip onto the flat counter electrode chip instead of forcing the device layer to be pressed down onto the counter electrode chip. On the other side, the flexible adhesive can suppress also external stresses working on the package which might affect also the distortion of the membrane. The role of adhesives for sensitive MEMS devices was discussed in detail in [18].

To conclude, stress within thin layers has a strong influence on the performance of a membrane-based focusing device. Even though soft tangential beams and soft adhesive bonding can shield and therefore reduce the influence of thin film stress, it is still a challenge to keep the stress contributions constant.

In respect to variations of process parameters and their influence on the optical performance of the final devices, the influence of the bonding process, the thickness and the thickness variation of the adhesive layer (the latter causing tilt of the device layer chip in respect to the counter electrode chip) have been identified as major influences on the devices’ performance. A typical spread of 30% for the sensitivity of the focusing devices (variation of focal length at a given supply voltage) has been found experimentally which can still be compensated by an according adjustment of the driving voltage because the pull-in voltage is increased by the ring-shaped electrode design in comparison to holohedral counter electrodes.

For applications, the long-term stability and especially the change of optical properties on large numbers of load cycles (electrostatic actuation cycles of the membrane) is important. Up to one million cycles, no systematic drift could be found. However, even more loads and long-term tests are needed, especially where the rim (influence of the adhesive) must be considered as well.

Most investigated devices are free from any measurable hysteresis. The characterization of dynamic focusing showed that dynamic focusing at frequencies of more than 1 kHz is possible and that the bandwidth of the actual devices is mainly limited by the used electronics (available current in the charging phase).

## Figures and Tables

**Figure 1 micromachines-12-00172-f001:**
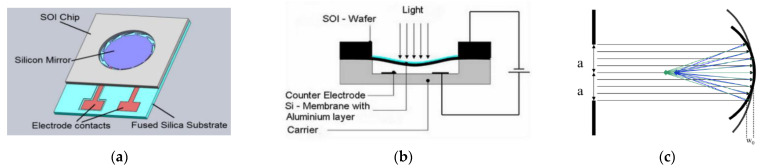
Schematic sketches of a membrane based focusing device. (**a**) The device consists of two chips bonded together: the silicon on insulator (SOI) chip containing the mirror membrane suspended with special formed beams at the rigid rim of the chip. The fused silica chip contains the counter electrode within a cavity etched into the fused silica; (**b**) cross section (not in scale): the membrane chip is bonded upside down to the fused silica chip. By using ring-shaped counter electrodes, the electrostatic force is acting on the outer perimeter of the membrane; (**c**) parallel incoming light is reflected back to a focal point defined by the parabolic amplitude, w_o_, of the electrostatically deformed membrane. In principle, an aluminum coating can be used to enhance reflectivity of the membrane surface [12,14].

**Figure 2 micromachines-12-00172-f002:**
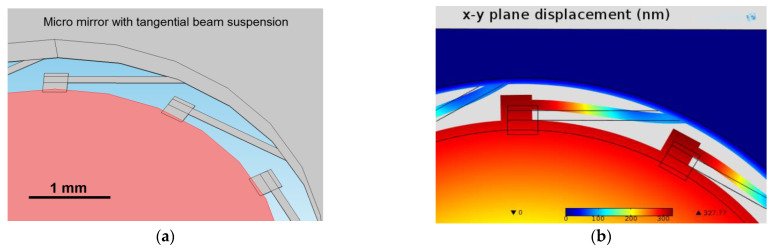
Special beam design to avoid out-of-plane distortion of the membrane. (**a**) Beam suspension of the membrane (red) to the outer rim (grey) for the tangential beam suspension design. (**b**) 3D FEM simulation of a membrane (6 mm diameter) suspended with tangential beams (width around 100 µm); membrane with a residual compressive stress of σ = −20 MPa (assumed for demonstration of the stress releasing function of the beams) showing the movement of membrane rim and the beams in the x-y plane only [12].

**Figure 3 micromachines-12-00172-f003:**
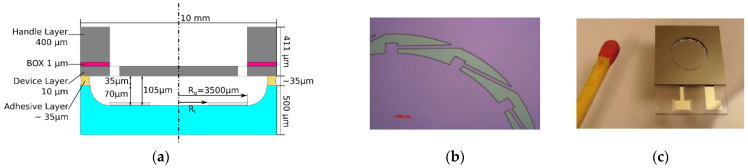
(**a**) Sketch showing typical dimensions and defining the inner radius R_i_ and the outer radius R_o_ for ring-shaped counter electrodes. (**b**) Microscopic picture of the manufactured micro mirror fixed by the tangential beam suspension. (**c**) Photo of a complete device before packaging in comparison to a match; both electrodes are electrically connected via the pads on the fused silica chip used for wire bonding.

**Figure 4 micromachines-12-00172-f004:**
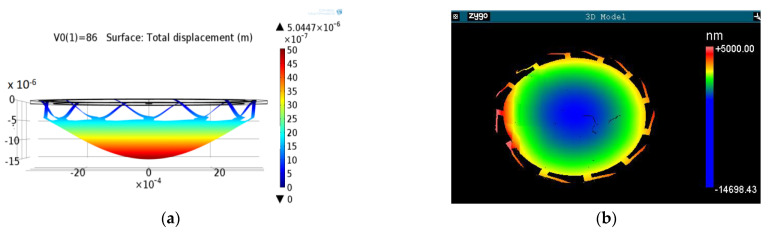
Electrostatically deformed membrane mirror (U_0_ = 86 V). (**a**): FEM simulation, membrane deformation w_o_ = 3.35 µm; (**b**): device measured with white light interferometer, w_o_ = 4.03 µm [16].

**Figure 5 micromachines-12-00172-f005:**
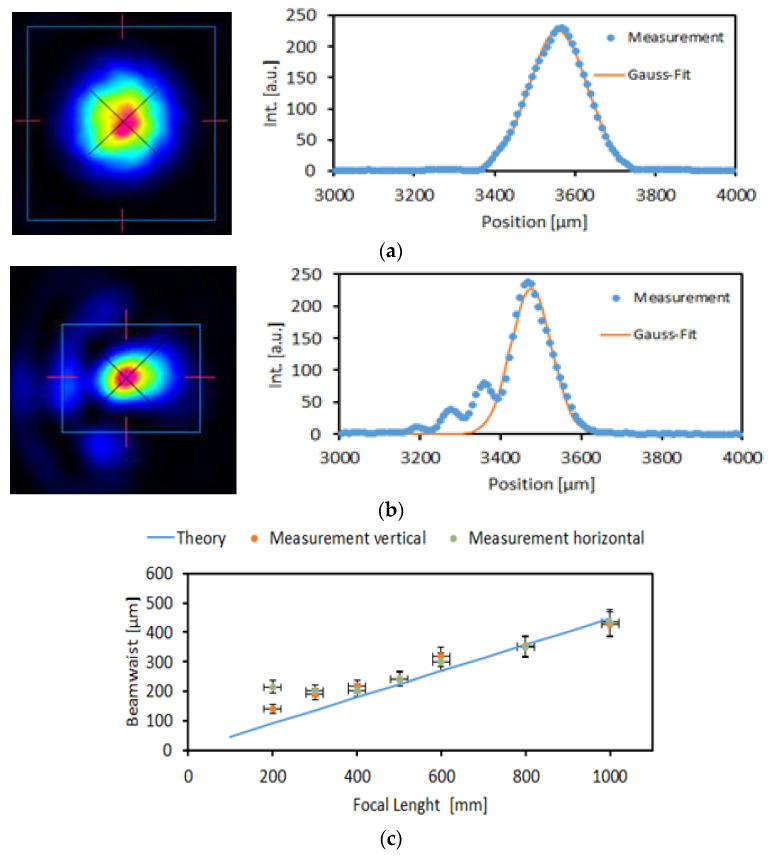
Characterization of imaging quality by focusing a laser spot with the FMM device (focal length f = 500 mm, U_0_ = 86 V) (**a**) for an aperture of 2 mm, (**b**) for an aperture of 5 mm. Shown are the measured 2D intensity distributions (left) and central scans through these distributions (right), (**c**) comparison between measured 1/e^2^—laser beam waist and a ZEMAX simulation of the beam waist assuming an ideal parabolic mirror as function of focal length [16].

**Figure 6 micromachines-12-00172-f006:**
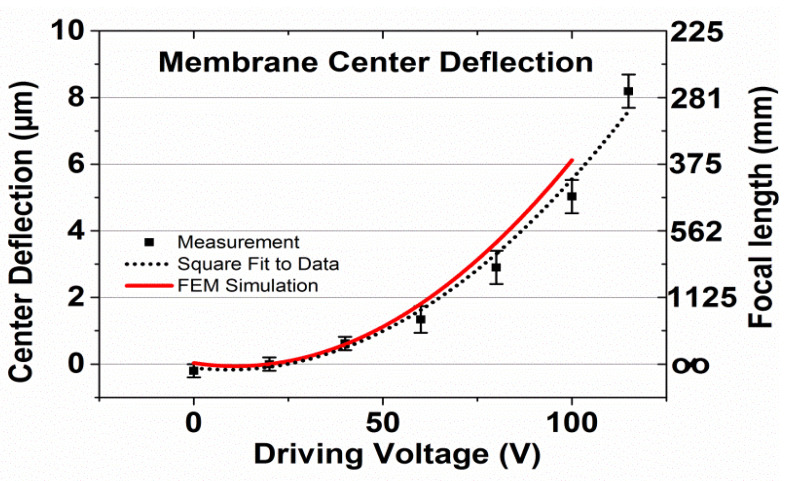
Center deflection w_0_ of the electrostatically deformed membrane and the according focal length f calculated using Equation (1) as a function of driving voltage. Membrane thickness 10 µm; membrane diameter 6 mm, holohedral electrode 2.9 mm radius; suspension beam length 1675 µm. Black squares: measured values; line: FEM simulation.

**Figure 7 micromachines-12-00172-f007:**
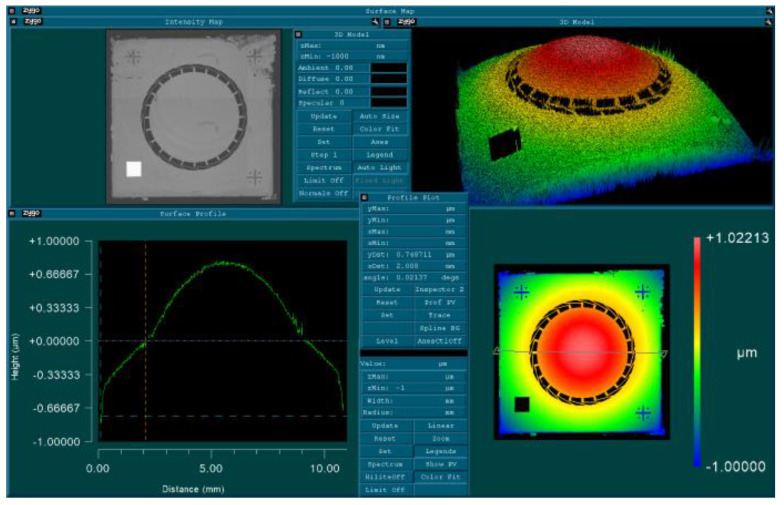
Distortion measurement over the full chip area of 10 × 10 mm^2^ (diameter of active mirror membrane: 6 mm, suspended at 24 tangential beams) demonstrating that not only the membrane shows a convex distortion but also the beams and even the thick chip rim are bending upward.

**Figure 8 micromachines-12-00172-f008:**
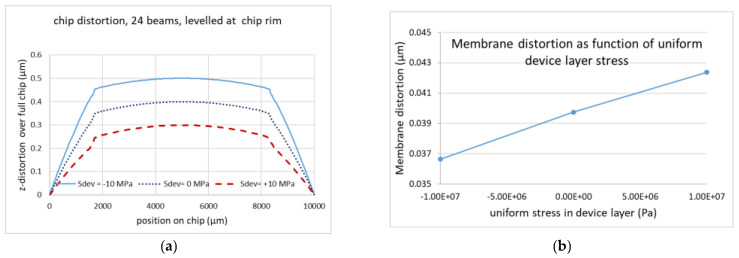
COMSOL simulation of a chip model with 24 tangential beams for different intrinsic stress values within the device layer (assumed uniform over the device layer thickness). (**a**) z-distortion as function of position on chip. The thick chip rim is warped upward; the thin membrane part shows a slight convex distortion of about 50 nm. (**b**) Maximum membrane distortion (z(at membrane center)-z(at membrane rim)) as a function of device layer stress. Chip width 10 mm, membrane diameter 6 mm, membrane thickness 10 µm. Different values of the intrinsic device layer stress are considered (−10 MPa, 0 M Pa and +10 MPa).

**Figure 9 micromachines-12-00172-f009:**
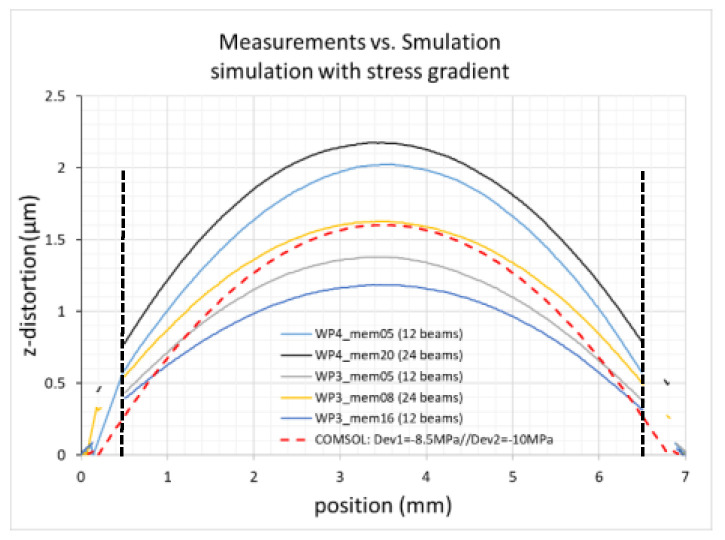
Evaluation of membrane distortion at U = 0 V. Shown are experimental results from 5 different devices made from 2 different wafers (mem05 and mem16: 12 tangential beams, mem08 and mem20: 24 tangential beams). As a comparison, a COMSOL simulation (red) for a model with 24 beams is shown where the device layer was split into two layers with different compressive stress values: a 9 µm thick layer with a residual stress of −10MPa and a 1 µm thick layer with −8.5 MPa; the dotted vertical lines mark the positions of the outer membrane rim.

**Figure 10 micromachines-12-00172-f010:**
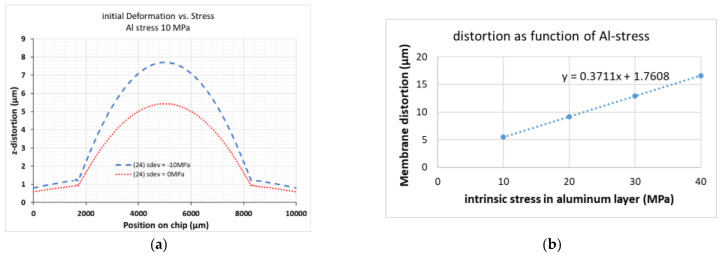
COMSOL simulation of the aluminum stress influence on the membrane distortion (difference between deflection at center and rim of membrane). (**a**) Tensile stress in an additional Al-layer of 10 MPa; this will introduce distortion even in the case of zero stress in the device layer (red line) and will significantly boost the distortion due to a compressive stress in the device layer (blue line). (**b**) Dependence of membrane distortion on Al stress for a device layer stress of −10 MPa with parameters used for a linear fit (dashed line); 24 beams, stress in the device layer −10 MPa.

**Figure 11 micromachines-12-00172-f011:**
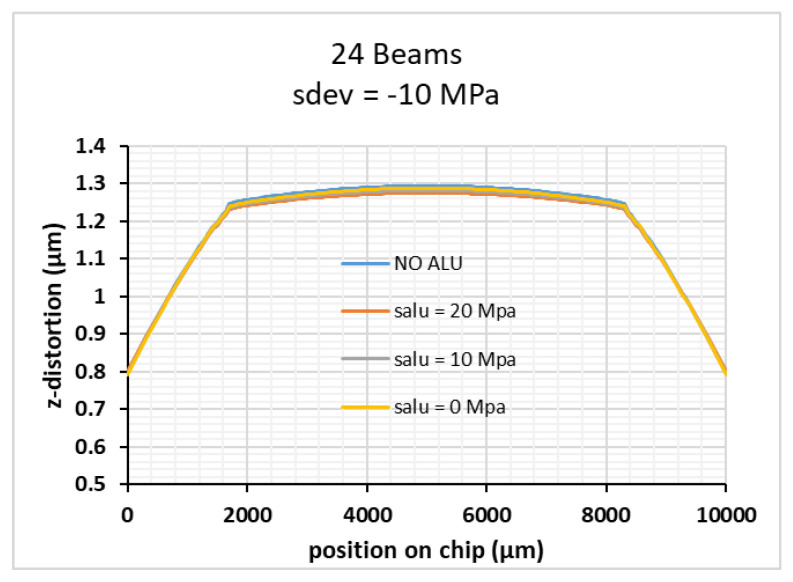
The large effect of Al stress can be compensated by deposition of an Al-layer with exactly the same stress values and thickness on the back side. In this case, the low membrane distortion without an aluminum reflective layer is recovered, i.e., the membrane part is almost flat.

**Figure 12 micromachines-12-00172-f012:**
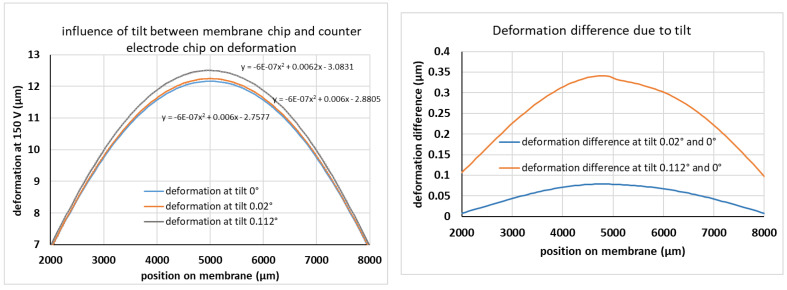
Simulated influence of typical tilt angles between electrode chip and device chip for a layout with 12 beams. The full chip was simulated, however only the relevant membrane deformation under a voltage of 150 V is shown. Left: membrane deformation at three different tilt angles. The equations show the parameters for a parabolic fit to the data for 0.112° (top), 0.02° (middle) and 0° (lowest equation). Right: simulated deformation differences at two tilt angles (0.02° and 0.112°) compared to the deformation with parallel electrodes (tilt 0°).

**Figure 13 micromachines-12-00172-f013:**
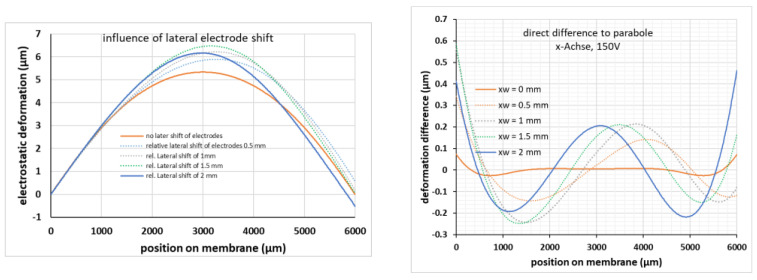
Influence of a relative lateral shift of the two electrodes. Model with 12 tangential, 100 µm wide beams, a ring-shaped counter electrode (R_i_ = 1.5 mm, R_o_ = 3.5 mm), a device layer stress of −2 MPa and an applied voltage of 150 V. Left: deformation for relative lateral shifts of electrodes in the range 0–2 mm. Right: deviation to individual parabolic fits.

**Figure 14 micromachines-12-00172-f014:**
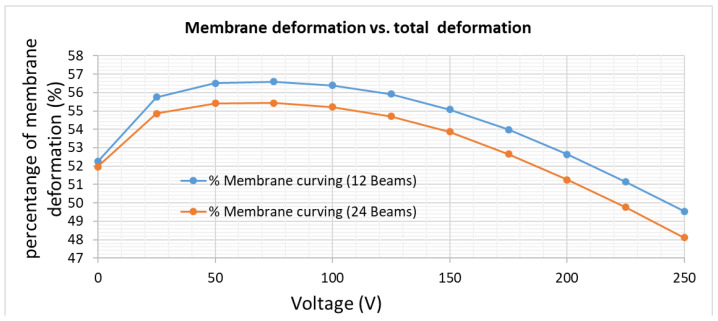
COMSOL simulation of the percentage of membrane deformation in relation to the total deformation including the beams; models with 12 and 24 beams and ring-shaped electrodes are used. The device layer was assumed with a uniform compressive stress of −2 MPa. The optically usable membrane deformation is only about 50% of the total deformation obtained.

**Figure 15 micromachines-12-00172-f015:**
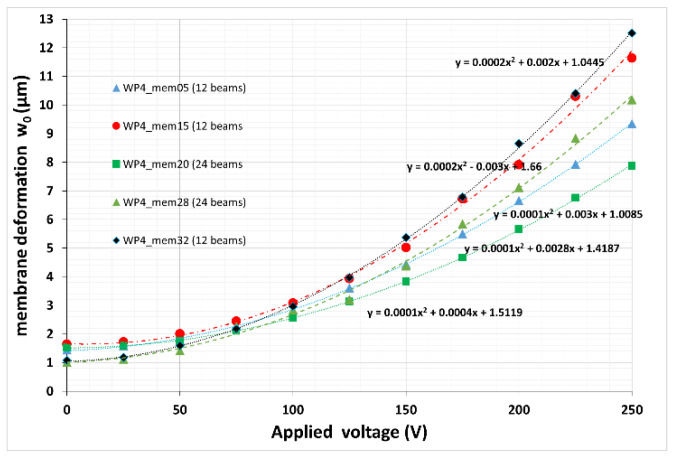
Measured voltage dependence of membrane deformation w_0_, which defines directly the focal length (Equation (1)). Layouts with 12 and 24 tangential beams were experimentally characterized. The lines are parabolic fits to the measured data with the given parameters.

**Figure 16 micromachines-12-00172-f016:**
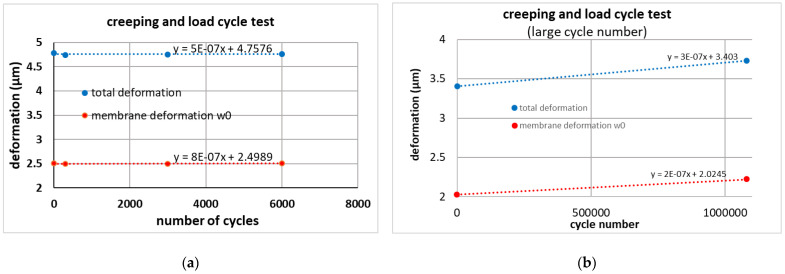
Creeping and load cycle test. (**a**) Between the shown deformation measurements, the devices were loaded with sinusoidal voltage cycles between 0 and 70 V (**b**) Deformation before and after about 1 Mio. cycles (0–70 V).

**Figure 17 micromachines-12-00172-f017:**
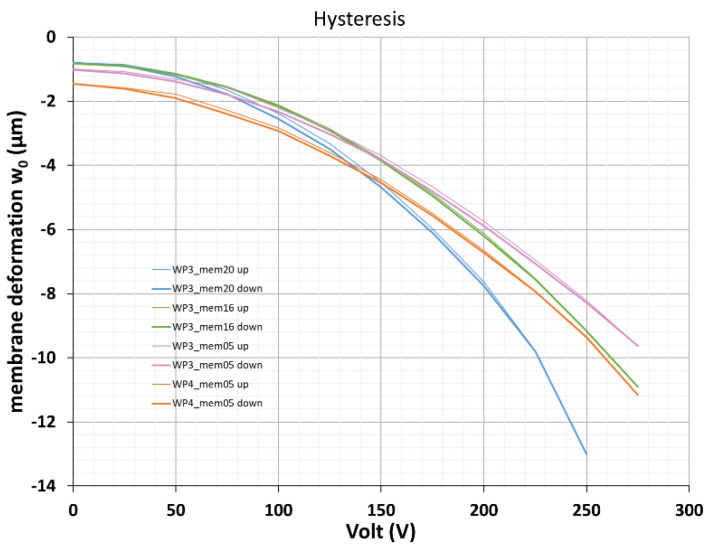
Hysteresis behavior of different devices. Four voltage cycles (upward and downward) for each device are shown.

**Figure 18 micromachines-12-00172-f018:**
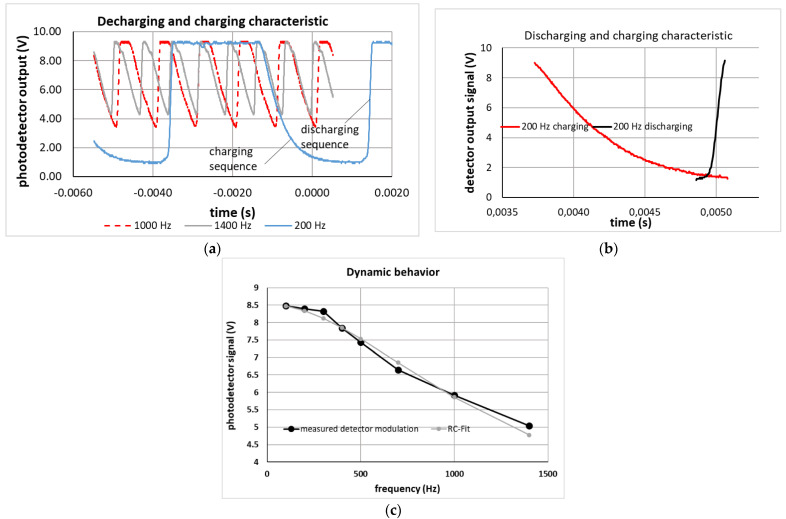
Characterization of dynamic behavior by focusing and defocusing a laser spot with the FMM device onto an iris diaphragm and measuring the light intensity behind the iris diaphragm. The membrane focusing device is powered with a square wave voltage function between 0 V and 170 V, (**a**) measured detector output at different frequencies, (**b**) charging and discharging sequence for modulation frequency of 200 Hz (time axes were shifted for overlap), (**c**) measured modulation amplitude as function of modulation frequency and fit with an RC-model (C = 2.8 × 10^−12^ F, R = 6 × 10^7^ Ω).

**Table 1 micromachines-12-00172-t001:** Properties of the used SOI wafers.

SOI Wafer Layer	Value
Device layer	P/Bor<100>; 10 µm
Buried oxide	1 µm
Handle layer	P/Bor<100>; 400 µm

**Table 2 micromachines-12-00172-t002:** COMSOL results for evaluation of voltage and electrode distance dependences. Membrane deflection w_o_ in µm.

Electrode Distance d (µm)	Max. Membrane Deformation w_0_ at U = 150 V	Max. Membrane Deformation w_0_ at U = 100 V	Max. Membrane Deformation w_0_ at U = 50 V	Max. Membrane Deformation w_0_ at U = 0 V
80	3.236	1.491	0.409	0.045
70	4.098	1.924	0.519	0.045
60	5.414	2.583	0.688	0.045
50	7.315	3.620	0.968	0.045
40	10.188	5.396	1.481	0.045

**Table 3 micromachines-12-00172-t003:** Creeping and stress cycle measurement results.

	Number of Cycles	Total Chip Deformation (µm)	Membrane Deformation (µm)
Run1	0	4.779	2.505
	300	4.737	2.490
	3000	4.755	2.497
	6000	4.763	2.506
Run2	0	3.402	2.024
	1,080,000	3.729	2.221

## Data Availability

Data available on request due to restrictions.

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
