# Peer review of "Evaluation and Optimization of a MOEMS Active Focusing Device"

_micromachines, 2021, doi:10.3390/mi12020172_

Round 1

Reviewer 1 Report

This manuscript presents a novel FMM device with simulation and experimental results. All procedures are well written and reasonable. The results are quite interesting and promising to future application for optical system. This reviewer recommends this article for publication after addressing following minor concerns.

Comments

  1. In figure 3, the authors used Ra and Ri. However, in the caption Ro and Ri were used.

  1. In section 3.2.2, electrode cavity is 70 um. But the gap between two layers is 60 um in Fig. 3. In addition, adhesive thickness of 35 um is not considered in the figure. It would be better to include adhesive layer in the figure. 35 um does not seem to be a negligible number.

  1. In section 3.2 the authors said that applying voltage is from 0 to 275V, but it seems that it was used up to 125V in figure 6. Is there any reason such as mismatch with simulation? (There are the results up to 250V in figure 15)

  1. It would be better to match the significant figures than to write all the numbers in table 2.

  1. There is a typo in line 10 on page 17 (i->in)

  1. For the practical uses, long term stability is important as the authors mentioned. How about the membrane deformation under DC voltage after long time actuation? Is there any decrease of deformation due to heat or other factors?

Author Response

First of all the authors would like to thank the reviewers for the comments and suggestions.

Authors’ Reply to the Review Report (Reviewer 1)

  1. In figure 3, the authors used Ra and Ri. However, in the caption Ro and Ri were used.

 Thanks, has been corrected in Fig. 3

  1. In section 3.2.2, electrode cavity is 70 um. But the gap between two layers is 60 um in Fig. 3. In addition, adhesive thickness of 35 um is not considered in the figure. It would be better to include adhesive layer in the figure. 35 um does not seem to be a negligible number.

 The adhesive has been included in Fig.3. Indeed it is not negligible here as we used a much thicker adhesive here than in previous work

  1. In section 3.2 the authors said that applying voltage is from 0 to 275V, but it seems that it was used up to 125V in figure 6. Is there any reason such as mismatch with simulation? (There are the results up to 250V in figure 15)

 As mentioned in the text, the results presented in Fig. 4-6 were already published. The according devices were using the same concept but a slightly different design and experimental conditions (e.g. counter electrode as mentioned in the text and thickness of the adhesive). Especially, the device presented in Fig.6 had a much smaller gap than those presented in the actual paper (thus these old ones are more sensitive to pull-in and therefore do not allow to apply high voltages). So the results in Fig. 6 and 15 cannot be compared directly. However, in respect to the electric field (the device is field driven, U²/d² (d: gap)) the results are corresponding to each other. Details of the layout for Fig. 6 can be found in [12].

  1. It would be better to match the significant figures than to write all the numbers in table 2.

The according dependence on d and U was investigated by simulation in detail in [14] (U. Mescheder, Z. Torok, W. Kronast, “Active focusing device based on MOEMS technology”, Proc. SPIE Vol. 6186, 618601 (2006), pp. 1-12). As a first estimation w0 will follow a (U²/d²) dependence as can be seen in the plots below. However, we thought that the slight differences can be better represented numerically. Especially because the dependence is different within the parameter space of U and d.. Therefore, out of Tab. 2, different parameters were compared to eq. (3) which clearly showed that the (U²/d²)-dependence of eq. (3) does not fit exactly with FEM results.  Further text was added to stress out the findings as well as how this was derived from Table 2.

Even though following approximately the dependence on U and d as in eq. (3), from the values in Table 2 different interesting findings can be derived for w0 within the two parameter space defined by U and d:

  • The voltage dependence of wo does not follow exactly a U² dependence: The relative deformation w0 (150 V)/wo(50 V) is only 7,9 for d= 80 µm and 6,9 for d = 40 µm (from eq.(3) 9,0 is expected). Assuming a Ux-dependence (i.e. neglecting any other terms) results in x=1,88 at d= 80 µm and x= 1,76 at d= 40 µm
  • Also the dependence of wo on d does not follow exactly a d-² dependence: The relative deformation w0 (80 µm)/wo(40 µm) is only 3,6 for U = 50 V and 3,1 for U = 150 V (from eq.(3) 4,0 is expected). Again assuming a dx dependence (i.e. neglecting any other terms) results in x=-1,63 at U= 150 V and x= -1,85 at U= 50 V.

Therefore, considering the variation of adhesive thickness as the dominating process influence we expect from the experimentally determined thickness variation of + 7 µm at a typical distance d of 70 µm (+10% distance variation) a corresponding variation of deformation w0 and thus focal length of the membrane focusing device of around + 17 % (average between x= -1.63 and x= -1.85) which can be compensated by an according adjustment of the driving voltage, especially as pull-in effects are considerably reduced by the ring-shaped electrode design. Even though the exact variation of w0 for a given distance variation will depend on the specific voltage U and the specific electrode distance d as can be derived from Table 3, the evaluation shows that a reasonable control of optical power is possible even for a large electrode distance variation of 10%.

in the enclosed document, the plots are shown

  1. There is a typo in line 10 on page 17 (i->in)

Thanks, we have tried also to correct some more typos throughout the text and polished some phrases.

  1. For the practical uses, long term stability is important as the authors mentioned. How about the membrane deformation under DC voltage after long time actuation? Is there any decrease of deformation due to heat or other factors?

We think that dynamic loads (changing membrane deformation) is the most critical situation, especially as large cycle number are expected at high frequency drive conditions and as a dynamically changing load might have an influence on the adhesive and thus on the sensitive membrane itself.

We did also a DC load for 3,5 hours where no change was found within the precision of the measurement (change of 33 nm of set deflection of 5.6 µm within that time), so much longer DC loads would be needed to quantify such (if any) small drift. Unfortunately, there is a practical limitation for a long term DC load (say over days or weeks) as a re-alignment in white light interferometer can cause larger measurement deviations than those we have to measure and keeping the probes in the white light microscope for such a long period would block this much used microscope for a long time. However, as mentioned on a short time frame (some h) we could not observe creeping under constant load within the measurement precision.

Remarks:

Besides polishing some phrases also some Figures were improved as some axes titles and legends were hard to read (to small).

Additionally to the previously marked changes we have added in 4.2.1 “influence of aluminium” information about the used models:

“In this first model including reflective coatings, the Al-layer was modelled as a 500 nm thick layer. To enable meshing, the 1µm thick BOX-Layer was split into two separate layers of 500 nm. This prevents meshing errors caused by unaligned block edges, and allows the COMSOL meshing algorithm to succeed……………..To prevent computation errors due to high aspect ratios of element dimensions during simulation of these very thin layers, the mesh either had to be very fine, which would increase computation time greatly, or the layers had to modelled with a different technique. To this end, we made use of the shell physics module offered by COMSOL, to model the thin aluminum layers not as 3D objects, but as a number of material properties and forces which are attached to the interfaces where the aluminum layers are in contact with the silicon membrane.”

In Fig. 10b we found a mistake in data extraction. For the Al-stress values 10 and 20 MPa, by lapse, the reference for the calculation of membrane distortion was not taken at the rim of the membrane but at the rim of the frame (where the beams are suspending the membrane). The corrected data show a much better (and expected) linear dependence on Al-stress and fit much better to the data at 30 and 40 MPa which were extracted correctly in the first submitted version

Reviewer 2 Report

In the presented paper Authors presents an evaluation of a MOEMS active focusing device which is realized by an electrostatically deformed thin silicon membrane.

All sections are prepared well - from introduction, trough design/simulation, fabrication, characterization to conclusions.

To be honest nothing to criticize or to be corrected.

Only on page 2 - what is HFU? – Hochschule Furtwangen University?

Author Response

Author's Reply to the Review Report (Reviewer 2)

In the presented paper Authors presents an evaluation of a MOEMS active focusing device

We have changed that paragraf, so HFU is not anymore used.

The results presented in this paper are based on the development of stress-free micromirrors with large apertures (5 mm) by using SOI wafers as the basis for creating large, free-standing membranes [11]. In that study the correlation between different membrane suspensions and the shape of the counter electrode was investigated, to encourage a parabolic deformation of the membrane itself.

Remarks:

Besides polishing some phrases also some Figures were improved as some axes titles and legends were hard to read (to small).

Additionally to the previously marked changes we have added in 4.2.1 “influence of aluminium” information about the used models:

“In this first model including reflective coatings, the Al-layer was modelled as a 500 nm thick layer. To enable meshing, the 1µm thick BOX-Layer was split into two separate layers of 500 nm. This prevents meshing errors caused by unaligned block edges, and allows the COMSOL meshing algorithm to succeed……………..To prevent computation errors due to high aspect ratios of element dimensions during simulation of these very thin layers, the mesh either had to be very fine, which would increase computation time greatly, or the layers had to modelled with a different technique. To this end, we made use of the shell physics module offered by COMSOL, to model the thin aluminum layers not as 3D objects, but as a number of material properties and forces which are attached to the interfaces where the aluminum layers are in contact with the silicon membrane.”

In Fig. 10b we found a mistake in data extraction. For the Al-stress values 10 and 20 MPa, by lapse, the reference for the calculation of membrane distortion was not taken at the rim of the membrane but at the rim of the frame (where the beams are suspending the membrane). The corrected data show a much better (and expected) linear dependence on Al-stress and fit much better to the data at 30 and 40 MPa which were extracted correctly in the first submitted version
